# Lion Hearts: Using the Intervention Mapping Framework to Develop a Family-Based CrossFit Program for Health Behavior Change

**DOI:** 10.3390/healthcare13233127

**Published:** 2025-12-01

**Authors:** Janette Watkins, Janelle Goss, Kelton Mehls, Deirdre Dlugonski, Danielle Symons Downs

**Affiliations:** 1Department of Kinesiology, College of Health and Human Development, The Pennsylvania State University, University Park, PA 16802, USA; jfg6171@psu.edu (J.G.); dsd11@psu.edu (D.S.D.); 2Department of Health and Human Development, University of Pittsburgh, Pittsburgh, PA 15261, USA; kdm139@pitt.edu; 3Department of Health & Clinical Sciences, College of Health Sciences, University of Kentucky, Lexington, KY 40536, USA; dee.dlugonski@uky.edu; 4Department of Obstetrics and Gynecology, College of Medicine, The Pennsylvania State University, University Park, PA 17033, USA

**Keywords:** Intervention Mapping, cardiovascular disease prevention, family-based intervention, community-based research, multigenerational health, implementation science

## Abstract

**Background/Objectives:** Physical inactivity and sedentary lifestyles remain leading behavioral risk factors for chronic disease across generations. Mothers with young children face unique barriers to exercise, including time constraints, fatigue, and limited access to supportive environments. *Lion Hearts* was developed to address these barriers through a family-centered, community-based approach that integrates physical activity, strength training, and health education. This protocol describes the systematic application of the Intervention Mapping (IM) framework to develop *Lion Hearts*, a multigenerational CrossFit-based program for mothers and children. **Methods:** Following the first four steps of the IM framework—needs assessment, matrices, intervention design, and program creation—behavioral determinants were identified through literature review, national data, and community input. The resulting 12-week program integrates twice-weekly family CrossFit sessions, monthly cardiovascular health workshops, and weekly home-based challenges delivered through local affiliates using a train-the-trainer model. **Results:** IM produced a theoretically grounded and evidence-based intervention targeting individual (self-efficacy, outcome expectations), interpersonal (modeling, relatedness), and environmental (access, social support) determinants. The process resulted in detailed logic models, behavior change matrices, and implementation materials, including family handbooks and coach guides. **Conclusions:**
*Lion Hearts* represents a scalable, multigenerational approach to CVD prevention that leverages existing community fitness infrastructure. By embedding prevention within family systems and CrossFit affiliates, the program offers a sustainable, replicable model to enhance physical activity, strengthen family health behaviors, and reduce intergenerational CVD risk.

## 1. Introduction

Cardiovascular disease (CVD) remains the leading cause of death among women in the United States, responsible for approximately one in every five female deaths [1]. The American Heart Association identifies motherhood as a vulnerable period that intensifies CVD risk factors, including reduced physical activity (PA) [2]. Maternal inactivity, obesity, and hypertension elevate women’s own CVD risk while simultaneously conferring intergenerational consequences: higher blood pressure, adverse lipid profiles, and lower fitness for their children [3,4]. Conversely, children who have physically active mothers mitigate these risk factors and are twice as likely to reach PA guidelines [5,6,7]. Addressing mothers’ PA levels during this life stage therefore represents an underutilized prevention opportunity with multigenerational implications [8].

Childhood represents a key developmental period for CVD prevention, as patterns of physical activity, fitness, and cardiometabolic health established early in life strongly influence risk trajectories into adulthood [9,10]. Consistent PA in this age group strongly predicts adult PA levels [11], yet fewer than one in four U.S. children consistently achieve the recommended 60 min of daily moderate-to-vigorous PA [12,13]. Mothers of young school-age children average some of the lowest levels of daily PA across the lifespan, with only 29.8% of U.S. mothers with pre-school children meeting the adult guideline of approximately 150 min/week of MVPA [14,15]. Likewise, researchers in the U.K. found that mothers with pre-school children achieved an average of 18 min/day of MVPA, while mothers with school-aged children averaged 26 min/day [16]. This inactivity heightens children’s risk for low activity levels, as maternal PA strongly predicts child PA through modeling and co-participation [17,18,19]. Despite this evidence, few interventions simultaneously engage mothers and children together in shared PA programming during this pivotal developmental stage [8,20].

Although aerobic PA receives the most attention in public health campaigns, muscle-strengthening activity remains especially low among women [21,22]. National surveillance data indicate that fewer than 26.9% of U.S. women meet the federal muscle-strengthening recommendation of two or more days per week, compared with roughly 35.2% of men [23]. This gap is even wider among mothers of young children, who cite fatigue, time constraints, and lack of access to supportive environments as primary barriers [24,25]. Strength-based exercise is critical for women’s cardiovascular and metabolic health, improving insulin sensitivity, body composition, and blood-pressure regulation while mitigating age-related muscle loss [26,27]. CrossFit offers an effective and scalable mode of strength training that combines aerobic and resistance exercise in time-efficient sessions [28,29]. These findings highlight a key opportunity to integrate strength training into maternal-focused prevention efforts.

Intergenerational interventions that engage parents and children together represent an especially powerful strategy for addressing sedentary behavior and promoting physical activity [30,31,32]. Such programs leverage reciprocal modeling and co-participation, where parents’ engagement enhances children’s motivation, and children’s enthusiasm reinforces parental adherence. This bidirectional influence fosters long-term behavior change and reduces sedentary time across generations, producing sustained improvements in both physical and psychosocial health. Family-centered interventions provide a promising but underused strategy to disrupt intergenerational CVD risk. Engaging mother–child dyads offers reciprocal motivation, accountability, and social support while fostering family cohesion [5,6,7]. However, existing programs are often resource-intensive, limited to clinical or school-based settings, and rarely engage children and mothers in non-aerobic-based activities such as skill and strength-based training [33]. This gap is critical, as early exposure to diverse, strength- and skill-based activities supports physical literacy, enhances body composition, and builds confidence for sustained engagement in PA [34,35].

Community fitness spaces—particularly CrossFit affiliates—represent a scalable, underused platform for family-centered prevention. CrossFit emphasizes functional movement, measurable progress [36], and social connection [37], features strongly associated with long-term adherence [37,38]. Affiliates provide accessible community hubs where mothers and children can participate together, overcoming common barriers to PA adherence such as cost, childcare, and social isolation. Specialty programs such as CrossFit Kids [39] offer developmentally appropriate training for children aged 4–11, making affiliates uniquely equipped to deliver safe, engaging, and scalable interventions for families [40,41]. Unlike traditional fitness centers such as the YMCA, which often separate adults and children through childcare services, CrossFit affiliates promote shared participation—allowing parents and children to train side by side—thereby strengthening family modeling, cohesion, and mutual motivation.

Despite decades of PA promotion research, few interventions target the mother–child dyad as the unit of change, with even fewer integrating cardiorespiratory fitness (CRF), body composition, and cardiometabolic biomarkers alongside behavioral outcomes as indicators of effectiveness [42,43,44]. Rigorous, theory-driven frameworks such as Intervention Mapping (IM) [45] are needed to bridge this gap and design interventions that are not only effective but also implementable in real-world community settings [20]. The *Lion Hearts* intervention applies IM to develop a community-based, multigenerational program for CVD prevention. *Lion Hearts* directly targets mother–child dyads, embedding prevention in CrossFit affiliates to create sustainable, socially supported change. Thus, the goals of this paper were to describe the IM process used to develop *Lion Hearts*, outline the application of IM theoretical foundations to *Lion Hearts*, and describe its utility as a replicable model for reducing intergenerational CVD risk.

*Lion Hearts* is a 12-week, family-centered CrossFit program designed to improve cardiovascular health and physical activity among mothers and their children. The intervention combines twice-weekly functional fitness sessions, monthly cardiovascular health workshops, and weekly home-based challenges delivered through local CrossFit affiliates. Rather than testing intervention efficacy, this paper focuses on the formative development phase of *Lion Hearts*, following the first four steps of the IM framework. The goal is to document how theory, evidence, and community input were systematically integrated to produce an evidence-based, family-centered program ready for future implementation and evaluation.

## 2. Materials and Methods

### 2.1. Overall Study Design

This study used the IM framework to guide the development of *Lion Hearts*, a multigenerational PA intervention for CVD prevention in mother–child dyads. IM integrates empirical evidence, behavioral theory, and implementation considerations to produce theoretically grounded, evidence-based programs suitable for real-world delivery. The first four IM steps (needs assessment, matrices, intervention design, program creation) were completed to develop and refine *Lion Hearts*, with implementation planning and evaluation (Steps 5 and 6) to follow in future phases. Figure 1 provides an outline of the IM steps applied in this study.

### 2.2. Research Context

*Lion Hearts* was designed in response to the need for family-centered CVD prevention strategies embracing the potential of community fitness settings—particularly CrossFit affiliates—as delivery platforms. The program was developed over a 3-month period through a collaborative process led by a multidisciplinary team with expertise in exercise science, health behavior change, CVD prevention, and community-based research. Team members met regularly to ensure consistent application of IM principles and to integrate diverse perspectives.

### 2.3. Planning Team Eligibility and Recruitment

The planning group consisted of faculty researchers (*n* = 4), community-based fitness professionals (*n* = 2), and mothers with lived experience in family-based health programs (*n* = 4). Advisory input was provided by local CrossFit affiliate owners (*n* = 2) who contributed expertise in functional training, class structure, and community engagement. Faculty researchers guided the theoretical alignment with Social Cognitive Theory, Self-Determination Theory, and Family Systems Theory, while graduate and undergraduate students supported literature reviews, data synthesis, and curriculum drafting. Mothers provided essential insights into family routines, barriers to participation, and motivational strategies for sustaining activity at home, ensuring that program content reflected real-world family dynamics. Fitness professionals and affiliate owners reviewed exercise selection, scaling options, and logistical considerations such as space, safety, and scheduling. This diverse composition ensured representation from both academic and community stakeholders, enabling the development of a program that is theoretically grounded, culturally relevant, and practically feasible for delivery in community fitness settings.

### 2.4. Theories, Models, and Frameworks for Implementation Strategies

Social Cognitive Theory (SCT) [46] informed strategies to enhance observational learning, self-efficacy, and outcome expectations within family contexts. The emphasis on reciprocal interactions among personal, behavioral, and environmental domains guided the selection of determinants and behavior change techniques. Self-Determination Theory (SDT) [47] provided the framework for designing activities that foster autonomy, competence, and relatedness to support intrinsic motivation, ensuring both mothers and children experienced meaningful engagement in health behaviors. Family Systems Theory (FST) [48] highlighted the importance of family dynamics, communication, and shared goal setting. This perspective informed strategies to strengthen family cohesion and sustain behavioral changes across generations.

### 2.5. Intervention Mapping (IM) Steps

IM guided the systematic development of the *Lion Hearts* program. Step 1 (Needs Assessment) involved creating a Logic Model of the Problem through a comprehensive problem analysis. This process included a literature review (2010–2025) of family-based physical activity and CVD prevention interventions, analysis of national surveillance data (CDC, NHANES, American Heart Association), and review of local community health assessments and facility inventories. This review synthesized key evidence on family-based physical activity, cardiovascular disease prevention, and intergenerational health promotion to identify behavioral determinants and environmental barriers relevant to mothers and children. In accordance with IM procedures, these findings are integrated within the Results Section rather than presented as a separate systematic review.

In Step 2 (Matrices), performance objectives were defined according to Intervention Mapping guidelines by specifying the concrete actions that mothers and children would need to perform to achieve the desired behavioral outcomes (e.g., adopting sustainable physical activity routines). The IM team, guided by Social Cognitive Theory, Self-Determination Theory, and Family Systems Theory, translated each overarching behavior into measurable, observable steps reflecting autonomy, competence, relatedness, and self-efficacy. Environmental outcomes were developed in parallel to identify the conditions within CrossFit affiliates necessary to support these behaviors. These included coach practices, facility adaptations, and community supports that foster modeling, accessibility, and consistent family participation. Both sets of objectives were finalized through consensus among academic researchers, community fitness professionals, and parent advisors to ensure ecological validity and alignment with real-world implementation contexts.

In Step 3 (Intervention Design), the IM team designed the *Lion Hearts* program, specifying frequency of sessions, integration of educational components, and strategies to overcome identified barriers by leveraging community facilitators through a CrossFit affiliate setting. Step 4 (Program Creation) undertook the development of materials to support a train-the-trainers model, including mother and child handbooks, coach implementation guides, and structured curricula to prepare CrossFit coaches to deliver the program effectively. All materials were designed to be evidence-based, visually engaging, culturally sensitive, and literacy-appropriate to ensure accessibility. Future steps (Steps 5 and 6) will employ implementation mapping within a community-based participatory research framework to refine components with stakeholder input, identify adoption and sustainability strategies, and evaluate both implementation outcomes (e.g., reach, fidelity, sustainability) and participant outcomes (e.g., PA, cardiovascular health, psychosocial well-being).

## 3. Results

The *Lion Hearts* development process successfully completed Steps 1–4 of the IM framework, resulting in a theoretically grounded, evidence-based intervention package designed for multigenerational CVD prevention in mother–child dyads. The Results Section summarizes the primary outputs generated from the first four steps of the Intervention Mapping (IM) framework. Because this paper describes the program development phase rather than implementation or evaluation, “results” refer to the key deliverables and decisions emerging from each step—namely, (1) the Logic Model of the Problem identifying determinants of inactivity among mothers and children, (2) the Logic Model of Change and behavior matrices translating these determinants into actionable objectives, (3) the design and structure of the 12-week *Lion Hearts* program, and (4) the creation of implementation materials including participant handbooks, coach guides, and child workbooks. These outputs collectively represent the evidence-based foundation for subsequent implementation and evaluation phases.

### 3.1. Step 1: Logic Model of the Problem

The needs assessment synthesized findings from national surveillance data, local community health reports, and an extensive literature review of family-based physical activity interventions. Analyses confirmed low adherence to physical activity guidelines in the target population, with only 23 percent of adults and 20 percent of children in the U.S. meeting PA recommendations [49]. Cardiovascular disease risk factors were prevalent in both groups, and barriers such as cost, transportation, and lack of family-friendly programming were consistently reported. The resulting Logic Model of the Problem identified determinants at individual (e.g., low self-efficacy, limited knowledge), interpersonal (e.g., lack of modeling and support), and environmental (e.g., limited access, affordability) levels (Figure 2).

### 3.2. Step 2: Logic Model of Change and Performance Objectives

Based on identified determinants, a Logic Model of Change was created to specify desired outcomes and performance objectives for both mothers and children. For mothers, these included leading and participating in family PA, setting shared health goals, and modeling healthy behaviors. For children, objectives included active participation in family sessions, engaging in independent activity, developing movement skills, and understanding the importance of cardiovascular health. Matrices of change objectives linked each performance objective to its corresponding determinants, providing a clear framework for selecting behavior change techniques (see Table 1 and Figure 3).

### 3.3. Step 3: Program Design

*Lion Hearts* will be structured as a 12-week program with three integrated components designed to support family health and engagement. Twice weekly, families will participate in 60 min CrossFit Sessions that incorporate functional movement, cardiovascular training, strength, and flexibility training activities scaled appropriately for mothers and children. Once a month, families will attend a 90 min Cardiovascular Health Education Workshop located at their local CrossFit affiliate, which will offer interactive learning on CVD risk, the benefits of physical activity, goal setting, and strategies for building supportive home environments. To extend the program beyond structured sessions, weekly Home-Based Family Challenges will encourage shared goals, reinforce skills, and promote family enjoyment. The overall design emphasizes developmental appropriateness, collaborative goal setting, and intrinsic motivation to build sustained engagement and long-term commitment to healthier lifestyles.

Unlike standard CrossFit or CrossFit Kids programming, *Lion Hearts* uniquely integrates family co-participation and cardiovascular education to promote both physical and relational health. Rather than focusing solely on performance or individual progression, the program emphasizes shared goal setting, maternal modeling, and home-based reinforcement to strengthen healthy habits within the family system. By merging the community-driven culture of CrossFit with evidence-based behavioral strategies, *Lion Hearts* represents a novel, family-centered approach to building cardiovascular fitness and lasting health behaviors across generations.

Each *Lion Hearts* session was designed to align with core CrossFit principles—functional movement, scalability, and community support—while incorporating behavioral strategies derived from Social Cognitive Theory and Self-Determination Theory. Families attend two 60 min CrossFit sessions per week, led by certified CrossFit coaches who have completed a *Lion Hearts* orientation on family-centered delivery, child safety, and motivational communication. Each session follows a consistent structure (see Table 2). Supervision is provided by a lead certified CrossFit coach, supported by trained assistants (e.g., service-learning students or staff) in a 1:6 coach-to-family ratio to maintain safety and engagement. Coaches receive ongoing mentorship from affiliate owners and the research team to ensure fidelity through checklists, observation forms, and monthly supervision calls.

### 3.4. Step 4: Program Production

Comprehensive program materials were developed to ensure consistency, accessibility, and family engagement across all components of *Lion Hearts*. Participant handbooks outline the structure of the 12-week program, including weekly schedules, progress-tracking tools, and practical tips for building active routines at home. Child-friendly workbooks incorporate age-appropriate illustrations, reflection prompts, and goal-tracking charts to support motivation and comprehension among younger participants. Coaching manuals provided by CrossFit offer standardized guidance on delivering CrossFit sessions, ensuring fidelity to safety, developmental appropriateness, and scalability for both mothers and children. The educational component of Lion Hearts includes monthly 90 min Cardiovascular Health Workshops and weekly Home-Based Family Challenges, each mapped to IM-identified behavioral determinants.

Workshop topics progress across the 12-week program to build cardiovascular literacy and self-regulatory skills (see Table 3). All content is included in the *Lion Hearts* family handbook and child workbook, designed with age-appropriate visuals, reflection prompts, and progress-tracking charts. Coaches receive a parallel implementation guide with step-by-step session outlines, cueing tips, and discussion questions to promote consistent delivery across affiliates.

### 3.5. Steps 5–6: Implementation Mapping

Consistent with IM guidelines, the next phase of *Lion Hearts* will involve implementation mapping to collaboratively refine and finalize delivery strategies, adoption supports, and evaluation procedures before testing the program in community settings. Implementation mapping will engage coaches, mothers, and affiliate owners to adapt materials, determine recruitment and retention strategies, and establish feasible assessment protocols. Following this participatory process, a mixed-methods evaluation will be conducted to assess feasibility, fidelity, and preliminary effectiveness across behavioral, physiological, and psychosocial domains. Table 4 outlines the anticipated health and implementation measures to be considered during this next phase. These indicators reflect the proposed framework that will be reviewed and adapted through implementation mapping prior to launch.

## 4. Discussion

This paper described the systematic application of the IM framework to develop *Lion Hearts*, a multigenerational, community-based physical activity intervention for CVD prevention. The findings of this formative research demonstrate how the structured application of the IM framework can translate theoretical constructs into a practical, community-based intervention. By explicitly connecting behavioral determinants identified in Step 1 to performance and environmental objectives (Step 2) and then operationalizing those objectives through program design (Step 3) and material production (Step 4), *Lion Hearts* provides a replicable model for developing multigenerational interventions that are both evidence-informed and implementation-ready.

The systematic needs assessment confirmed that low PA levels and high CVD risk factors are prevalent among mothers and children and that barriers such as cost, scheduling conflicts, and lack of family-friendly facilities limit engagement in PA. The Logic Model of the Problem and Logic Model of Change developed through IM provided a clear blueprint for addressing these determinants. Program design and material production emphasized scalability, developmental appropriateness, and family-centered engagement, positioning *Lion Hearts* for successful implementation in community fitness facilities such as CrossFit affiliates.

A unique strength of *Lion Hearts* is its scaling potential through existing infrastructure. CrossFit affiliates—numbering more than 10,000 across 150 countries [50]—provide a ready-made, community-based delivery system for family-centered prevention. Unlike traditional fitness centers, affiliates emphasize functional, scalable programming [36] and incorporate movements such as lifting, squatting, and carrying. These movements translate directly to the physical demands of daily motherhood and help to reduce musculoskeletal injury risk and enhance strength and cardiorespiratory fitness [29,51,52], which are key physiological predictors of CVD morbidity and mortality [53]. Additionally, many coaches pursue certifications such as CrossFit Kids [39], equipping them to deliver age-appropriate programming for children while simultaneously engaging adults. This dual capacity allows mothers and children to train side by side, embedding role modeling and shared routines while reducing barriers related to childcare, cost, and scheduling. By leveraging a workforce and infrastructure that already exists, *Lion Hearts* bypasses the resource-intensive process of building new delivery systems, offering a replicable model that embeds cardiovascular prevention directly within trusted community hubs.

Despite the promise of leveraging CrossFit affiliates as community delivery platforms, several considerations affect the feasibility and accessibility of *Lion Hearts* and similar programs. First, program costs—including facility access, coach training, and participant fees—can limit scalability in underserved or rural areas. To mitigate these challenges, *Lion Hearts* incorporates a train-the-trainer model, allowing existing coaches to be trained in family-centered delivery with minimal added expense and no new infrastructure. Additionally, partnerships with schools, community centers, and local sponsors can offset participation costs and increase reach among families who may not traditionally access CrossFit facilities. Geographic availability also presents a barrier, as CrossFit affiliates are unevenly distributed across communities. Integrating program delivery into shared-use spaces and leveraging mobile or pop-up formats may help address this limitation. Finally, accessibility for families of varying fitness levels and abilities remains essential. To enhance inclusivity, *Lion Hearts* emphasizes movement scaling, supportive coaching, and culturally responsive materials to ensure safe, developmentally appropriate participation. These considerations will be systematically examined during future implementation mapping and pilot testing to inform sustainable scale-up.

Although the IM framework has been widely used to develop family- and school-based health interventions, *Lion Hearts* advances this body of work by extending IM into a novel community fitness context. Previous IM-based family interventions have primarily targeted diet or general lifestyle education within clinical or educational settings, often emphasizing aerobic activity or parental guidance rather than co-participation [54,55,56]. In contrast, *Lion Hearts* applies IM to design a multigenerational, strength-based program delivered through existing CrossFit affiliates. This approach expands the ecological scope of IM by integrating behavioral science with community fitness infrastructure, thereby enhancing scalability and accessibility.

This development phase produced a comprehensive intervention package; however, implementation planning and evaluation design (Steps 5 and 6 of IM) remain future work. These steps will be undertaken through implementation mapping guided by a Community-Based Participatory Research (CBPR) approach. By actively involving mothers, youth, fitness coaches, and community partners as equal collaborators, CBPR will ensure that adoption, delivery, and evaluation strategies are contextually relevant, culturally appropriate, and sustainable [57,58]. This participatory process will enhance local ownership and capacity, increasing the likelihood that *Lion Hearts* can be maintained and scaled after the research phase ends. Framing Steps 5 and 6 as a collaborative process acknowledges that even the most theoretically sound and evidence-based programs can falter without alignment to community realities [20]. Implementation mapping with CBPR will allow the intervention to be adapted for logistical feasibility, integrate local insights into recruitment and retention strategies, and establish realistic evaluation metrics that capture both health outcomes and implementation success.

The intergenerational design of *Lion Hearts* addresses sedentary behavior at both individual and family levels by aligning motivation, accountability, and environmental support across generations. By encouraging mothers and children to move together, the program transforms daily routines into shared opportunities for physical activity, strengthening family cohesion while promoting lifelong movement habits. Future research will focus on co-developing implementation protocols, training and supporting delivery staff, and establishing robust evaluation procedures to assess both proximal health changes (e.g., PA levels, cardiovascular indicators) and long-term maintenance. Rigorous mixed-methods evaluation will also explore mechanisms of change and contextual factors influencing outcomes. In this way, *Lion Hearts* lays the groundwork for a future community-owned program that may contribute to healthier behaviors and reductions in cardiovascular risk across generations once implemented and empirically evaluated.

## Figures and Tables

**Figure 1 healthcare-13-03127-f001:**
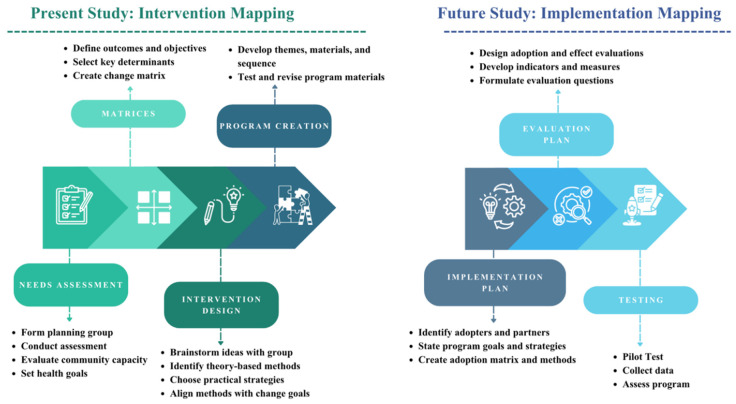
Study Steps Outline. Overview of the Intervention Mapping process applied to the Lion Hearts program. The “present study” encompasses Steps 1–4 (Needs Assessment, Matrices, Intervention Design, and Program Creation), describing the development of Lion Hearts. “Future studies” will address Steps 5–6 (Implementation Planning and Evaluation) to refine, implement, and assess the intervention in community settings.

**Figure 2 healthcare-13-03127-f002:**
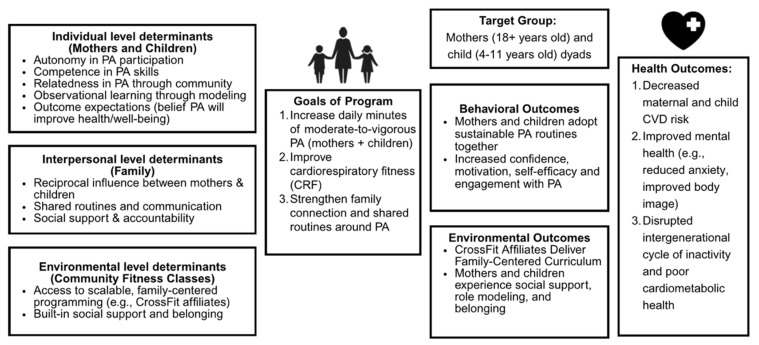
Logic Model of the Problem. The Logic Model of the Problem illustrates how individual, interpersonal, and environmental determinants contribute to low physical activity among mothers and children. These multilevel factors inform the Lion Hearts program goals and expected behavioral and health outcomes, including increased family activity, improved fitness, and reduced intergenerational risk for inactivity and poor cardiometabolic health.

**Figure 3 healthcare-13-03127-f003:**
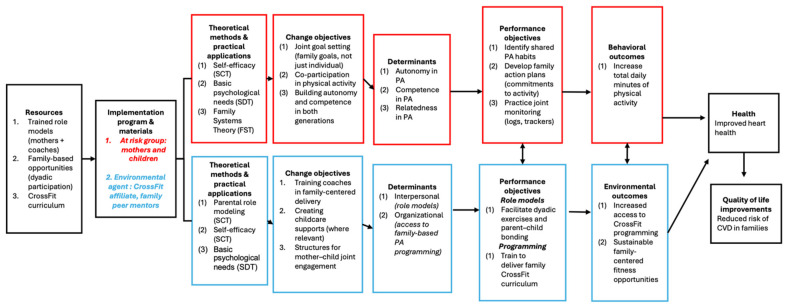
Lion Hearts Implementation Mapping Logic Model. Logic model outlining the Lion Hearts two-generation implementation strategy. The model maps resources, theoretical methods, change objectives, determinants, performance objectives, and expected behavioral and environmental outcomes leading to improved physical activity and cardiometabolic health among mothers and children in community CrossFit settings.

**Table 1 healthcare-13-03127-t001:** Logic Model of Change and Performance Objectives.

Behavioral Outcome 1: Mothers and Children Adopt Sustainable PA Routines Together				
Performance Objective	Autonomy	Competence	Relatedness	Self-Efficacy/Modeling (SCT)
PO1. Identify opportunities for PA together	Mothers/children choose activities they enjoy	Knowing safe, accessible activities and how to initiate them	Identifying shared interests	Mothers confident in planning PA and modeling it for children
PO2. Establish family routines for PA	Choosing times/locations that work for family	Demonstrating knowledge of simple, scalable PA (e.g., functional fitness)	Negotiating shared routines	Mothers modeling consistent PA habits
PO3. Engage in PA with family/peers	Choosing to join a group/family PA	Demonstrating skills to participate together	Strengthening peer/family connection through shared activities	Building confidence through practice and role modeling
PO4. Commit to sustained PA	Choosing ongoing involvement	Recognizing fitness progress	Maintaining supportive relationships	Mothers reinforcing positive expectations and outcomes
Behavioral Outcome 2: Increased Confidence and Engagement with PA				
Performance Objective	Autonomy	Competence	Relatedness	Self-Efficacy/Modeling (SCT)
PO1. Recognize personal preferences and motivators for PA	Participants independently choose the types of activities they are interested in.	Learn to assess what kinds of PAs feel good, are feasible, and match their fitness level.	Share preferences and motivations within family or peer groups.	Mothers/peers model enthusiasm for selected activities, reinforcing that “there is no one right way.”
PO2. Experiment with different PA forms to find “fit”	Choose to try new or varied physical activities.	Acquire basic skills or techniques in those new activities (e.g., walking, dance, cycling, bodyweight circuits).	Engage together (family, friends, groups) in trial sessions of different PA types.	Mothers or peer “buddies” model trying new PA and share learning from trial and error.
PO3. Set and adjust individualized PA goals	Autonomously set goals that feel challenging but achievable.	Learn how to break larger goals into smaller steps; monitor progress.	Share goal setting and progress with a supportive network.	Mothers model self-regulatory behaviors (tracking, adapting goals when needed) and verbalize coping with setbacks.
PO4. Regularly participate in PA and expand engagement	Make ongoing choices to schedule and attend PA sessions.	Develop increasing proficiency, stamina, and variety in movement.	Deepen connection by exercising together or participating in group/community PA.	Mothers model persistence, reinforce success, and encourage overcoming barriers.
Environmental Outcome 1: CrossFit Affiliates Deliver Family-Centered Curriculum				
Performance Objective	Role Modeling (FST/SCT)		Family and Environmental Reinforcement (FST/SCT)	
PO1. Identify and align local opportunities for family PA	Coaches and affiliate leaders model proactive engagement—identifying community resources and demonstrating initiative in creating family-inclusive spaces.		Affiliates and families collaborate to reinforce shared goals for accessible PA environments, shaping collective norms that value family involvement.	
PO2. Promote and communicate family-friendly programming	Coaches and mothers model enthusiasm and inclusivity when presenting family sessions, highlighting that PA is a shared experience.		Affiliates reinforce participation through environmental cues (e.g., posters, success stories) and social reinforcement from peers and families.	
PO3. Ensure accessible equipment and supportive spaces	Coaches model equitable use and care of equipment, showing adaptability in scaling for different ages and abilities.		Families and coaches co-create supportive settings—reinforcing expectations that all members can participate safely and effectively.	
PO4. Establish consistent, predictable scheduling	Coaches model reliability and consistency, demonstrating that structured routines help sustain engagement.		Families and affiliates co-develop schedules and reinforcement systems (e.g., check-ins, progress tracking) that sustain family attendance.	
Environmental Outcome 2: Mothers and Children Experience Social Support, Role Modeling, and Belonging				
Performance Objective	Environmental Determinants			
PO1. Recruit and empower positive role models	Affiliates identify and engage coaches, mothers, and peer mentors who model supportive, inclusive, and family-centered physical activity behaviors.			
PO2. Train and equip role models to deliver the family-centered curriculum	Affiliates provide structured onboarding and ongoing mentorship for coaches, service-learning students, and assistants to ensure fidelity, empathy, and developmental appropriateness in delivery.			
PO3. Integrate role models into family sessions and community networks	Affiliates establish systems that embed role models in family classes, peer circles, and community partnerships to promote consistent social reinforcement and shared accountability.			
PO4. Evaluate and celebrate social support and belonging	Affiliates monitor participation, gather family feedback, and publicly recognize role models and families to strengthen community connection and sustain engagement.			

**Table 2 healthcare-13-03127-t002:** Overview of Lion Hearts Weekly CrossFit Sessions (Step 3: Program Design).

Component	Description	Duration/Frequency	Lead Personnel and Supervision	Behavioral Targets (IM Constructs)
Welcome and Goal Review	Families review previous week’s progress and set new goals collaboratively. Coaches prompt reflection on intrinsic motivation and self-monitoring.	5–10 min, at start of each session	Certified CrossFit coach and assistant; 1:6 coach-to-family ratio	Autonomy, self-efficacy, self-regulation
Warm-Up and Skill Development	Introduction and scaling of fundamental functional movements (e.g., squats, carries, balance drills) tailored for mothers and children.	10–15 min	Lead coach models proper technique; assistants ensure safety	Competence, modeling, confidence
Workout of the Day (WOD)	Family participation workout combining strength, endurance, and agility (e.g., partner AMRAPs, relay circuits). Movements scaled by age and ability.	20–25 min, twice weekly	Supervised by certified CrossFit coach trained in family-centered delivery	Relatedness, mastery experience, enjoyment
Cool Down and Reflection	Guided discussion and stretching period to reinforce teamwork, effort, and positive feedback.	10–15 min	Coach-led reflection with family discussion prompts	Relatedness, positive reinforcement, competence
Supervision Model	Sessions overseen by certified CrossFit coaches with *Lion Hearts* orientation; ongoing fidelity monitoring through checklists and monthly mentor calls.	Continuous	Research team and affiliate owner provide oversight	Fidelity, sustainability, implementation quality

**Table 3 healthcare-13-03127-t003:** Cardiovascular Health Workshops and Home-Based Family Challenges.

Component	Content Example	Timing/Frequency	Delivery Personnel	Behavioral Focus (IM Constructs)
Workshop 1—Understanding Heart Health	Interactive demonstrations on heart rate, blood pressure, and recognizing exertion. Families measure their pulse and discuss intensity zones.	Week 1 (90 min)	Researcher and coach co-facilitation	Knowledge, outcome expectations
Workshop 2—Fueling Movement	Education on balanced nutrition, hydration, and family meal planning to support activity.	Week 4 (90 min)	Registered dietitian or trained coach	Environmental support, self-efficacy
Workshop 3—Goal Setting and Stress Management	SMART goal training, stress-coping techniques, and strategies to overcome barriers to PA.	Week 8 (90 min)	Researcher or behavioral specialist	Self-regulation, problem solving
Workshop 4—Celebrating Progress	Family reflection on achievements, long-term planning for maintenance, and group celebration.	Week 12 (90 min)	Coaches and research team	Reinforcement, sustainability
Weekly Family Challenges	Examples include: Family Step Challenge, Strength Circuit Saturday, and Active Kindness activity. Each challenge reinforces PA and family bonding.	Weekly, self-paced	Families with remote coach support	Autonomy, modeling, social support
Support Materials	Family handbooks, child workbooks, and coach guides include visual instructions, reflection prompts, and tracking charts.	Throughout 12 weeks	Research and coaching team	Accessibility, engagement, consistency

**Table 4 healthcare-13-03127-t004:** Anticipated Health and Implementation Measured for Implementation Mapping Phase.

Domain	Measure/Indicator	Description and Method	Purpose/Rationale
Cardiorespiratory Fitness (CRF)	Graded bike erg test (mothers); 20 m shuttle run (children)	Proposed standardized field tests conducted pre- and post-program; optional heart rate monitoring with wearable sensors.	Evaluate changes in cardiovascular endurance and aerobic capacity.
Physical Activity and Sedentary Behavior	Accelerometry (ActiGraph or Fitbit)	Proposed continuous wear over seven days at pre-, mid-, and post-intervention to assess MVPA, step counts, and sedentary time.	Quantify objective changes in daily activity and inactivity patterns.
Body Composition	Height, weight, BMI, waist circumference, bioelectrical impedance (optional)	Standardized anthropometric and composition data collected by trained personnel.	Assess body composition changes associated with improved cardiovascular health.
Blood Pressure and Resting Heart Rate	Automated sphygmomanometer (average of two seated readings)	Measurements taken after five minutes of rest at baseline and post-program.	Monitor cardiovascular risk indicators and physiological responses.
Behavioral and Psychosocial Indicators	Self-efficacy for PA, enjoyment, family cohesion, parental modeling, workshop satisfaction	Validated questionnaires administered pre- and post-program to capture psychosocial and motivational mechanisms.	Examine behavioral mediators influencing engagement and adherence.
Implementation Metrics	Attendance, retention, coach fidelity checklists, participant feedback	Logs, fidelity assessments, and interviews collected during delivery; finalized through implementation mapping.	Evaluate feasibility, reach, and sustainability of program delivery.

## Data Availability

The original contributions presented in this study are included in the article. Further inquiries can be directed to the corresponding author.

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
