# Peer review of "Lion Hearts: Using the Intervention Mapping Framework to Develop a Family-Based CrossFit Program for Health Behavior Change"

_healthcare, 2025, doi:10.3390/healthcare13233127_

Round 1

Reviewer 1 Report

Comments and Suggestions for Authors

Dear authors,

Thank you for the opportunity to review this manuscript. Please find my comments below.

Introduction

Line 50: “Children represent a critical window for CVD prevention.” I do not understand what the authors mean by this sentence.

I believe that the authors should emphasize the importance/benefits of intergenerational interventions to combat sedentary behavior and increase PA.

I also recommend including a brief sentence that explains exactly what the Lion Hearts program is, to facilitate the reader’s experience. Although this is mentioned at the end of the Introduction, it could be explained in a clearer and more straightforward manner.

Methods

In Figure 1, what do “present” and “future” studies mean?

It is not clear what was done. This paper aimed to describe the IM process used to develop Lion Hearts; however, there is no concrete information on how the process was developed. I assume that you recruited participants (who?), developed some type of activities (which?), and evaluated them (how?). Without this information is difficult to understand and interpret the results.

Moreover, the authors mention a literature review (2010-2025), but it is unclear whether the results are presented in this study.

How were the performance objectives defined? The same question applies to environmental outcomes.

From my perspective, the paper lacks clear information to support the readers in understanding the results and further discussion.

Author Response

See attached, please

Reviewer 2 Report

Comments and Suggestions for Authors

Thank you for the opportunity to review this manuscript. The authors present the development of a novel intervention, Lion Hearts, which utilises the Intervention Mapping framework to design a physical activity programme targeting mothers and their children.

This manuscript does not address a specific research question per se, as its primary aim is to describe the development process of the Lion Hearts intervention using the IM framework. Nevertheless, the authors provide a comprehensive and well-structured account of the application of IM, with clear justification for each step and relevant theoretical underpinnings.

The topic is both original and highly relevant. The manuscript addresses a notable gap in the literature by focusing on family-based physical activity interventions, specifically targeting mothers with young children; a demographic known to have particularly low physical activity levels due to time constraints, fatigue, and caregiving responsibilities.

The use of CrossFit as the intervention modality is innovative. It combines aerobic and strength training, which is particularly important given the well-documented underrepresentation of strength training among women. This dual-modality approach is a strength of the intervention and aligns well with current public health recommendations.

Furthermore, the intervention’s emphasis on co-participation between mothers and children offers a promising strategy to overcome common barriers to physical activity and to promote intergenerational health benefits.

This manuscript contributes meaningfully to the growing body of research on family-based physical activity interventions. Currently there is very limited research on family-based physical activity interventions. Of the limited family-based physical activity interventions that have been evaluated to date, the focus has been on child physical activity levels, omitting the potential for such interventions to support child and parent health outcomes; something highlighted in a review by Kriemler et al (2011). A recent paper which explored a father and daughter physical activity intervention (that included co-participation in fun games, challenges and sports) reported increased father and daughter physical activity levels and reduced sedentary time Morgan et al (2019). However, whilst this research is interesting and informative, there is still an evident evidence gap with regard to co-participation in physical activity within the family-unit, in mothers specifically, and whether these benefits extend to enhancing health factors such as cardiometabolic health. Thus, the intervention described by the authors in the present manuscript addresses a critical gap in the literature by including strength-based training and using of Cross-Fit as a delivery platform, which further distinguish this work from existing interventions.

It would be nice to have more detail about steps 3 and 4. Steps 1 and 2 have been described and justified in great detail, however steps 3 and 4 provide very little detail. It would be handy to have some examples about how those 60-minute sessions will run. For example, will mothers and children be doing the same session and, if so, will someone be assisting the mothers in supervising their children so that the mothers can focus on their own workout and not just focusing on what their child is doing. It would be helpful to perhaps give an example of one of the sessions in the supplementary material. This also extends to the Cardiovascular Health Education Workshops and weekly Home-Based Family Challenges-what will theses actually include, and how will they be structured. Additionally in step 4 some examples of what will be included in the participant handbooks (as supplementary material) would be useful.

Furthermore, while I appreciate this is only about steps 1-4, more information about what ‘health markers’ will be included and how these will be assessed in steps 5/6 would be helpful, especially given the emphasis placed on CVD health in the introduction and throughout the manuscript.

This is a methodology paper, and therefore there are no conclusions. However, in the discussion the authors have summarised the methods and results well and have justified the need for such an intervention and why such an intervention requires very careful development and thus why the IM framework is the most appropriate method for designing this intervention.

The references cited throughout the manuscript are appropriate and there is a sufficient number of references used in appropriate contexts.

Tables could be formatted a little better to avoid cells being split between two pages- it might be better to do a split page and put the tables on landscape pages.

Personally, I think it would look better if you had a paragraph space between the final line of text and the figure/ table title- it would just make it look a little tidier. Having it as it is at the moment makes everything feel very bunched in and a little untidy.

Typographical errors:

  1. Formatting inconsistency: There is inconsistency in the formatting of performance objective identifiers. In the table under ‘Behavioural Outcome 1’, identifiers are written as “PO.1”, whereas in subsequent tables (‘Behavioural Outcome 2’ and ‘Environmental Outcomes’), the format “PO1.” is used. Please standardise this formatting throughout the manuscript.
  2. Abbreviation usage: Please ensure that all abbreviations are defined upon first use. For example, on line 270, the term “CBPR” is used without prior definition. It would be helpful to introduce this as “Community-Based Participatory Research (CBPR)” before using the abbreviation.

Author Response

See attached, please

Reviewer 3 Report

Comments and Suggestions for Authors

The manuscript does not include any data collection, intervention implementation, or empirical results. Although the authors have designated the article type as “Article,” based on its current structure and content, it should be classified as a “Protocol.” Therefore, I recommend changing the article type accordingly.

Title: The title emphasizes “Cardiovascular Health,” which suggests that CrossFit is used exclusively for cardiovascular improvement. However, CrossFit has broader applications that contribute to multiple aspects of health and physical development. The title should be revised to reflect the more comprehensive and behavioral scope of the intervention.

Abstract: Similarly, the abstract begins with a statement about cardiovascular disease (CVD), which creates the impression that this is a clinical prevention study. Yet, no cardiovascular or physiological indicators are measured. It would be more appropriate to highlight the behavioral and community-based focus of the program rather than CVD prevention per se.

Line 22–24 could be revised to remove the dash and improve readability. The dash should be removed. "Methods: The study followed the first four steps of the Intervention Mapping framework—namely, needs assessment, creation of matrices, intervention design, and program development—to identify behavioral determinants through literature review, national data, and community feedback."

Measurement concerns: There are no physiological or clinical markers (e.g., blood pressure, resting HR, HRR, VOâ‚‚max, lipid profile, BMI, waist circumference) or medical monitoring described. Only behavioral constructs (participation, self-efficacy, family routines) are defined. Thus, it is unclear how CVD prevention or reduction will be evaluated.

Line 286–288: The sentence “In this way, Lion Hearts will progress from a well-designed intervention concept to a fully tested, community-owned program with strong potential for impact on reducing CVD risk across generations” appears overly assertive given that no empirical data or testing has been conducted. The claim should be moderated to align with the descriptive and developmental nature of the manuscript.

Scope and novelty: The study presents Steps 1–4 of the Intervention Mapping framework (planning and design) but does not include Steps 5–6 (implementation and evaluation). The use of the IM framework is already common in public health, exercise, and behavioral science research. The authors should clearly discuss how their work differs from or extends previous IM-based family interventions, emphasizing its unique contribution.

Discussion: The discussion is predominantly positive and lacks critical reflection. Issues such as feasibility, cost, access to CrossFit facilities, and sustainability among low-income or under-resourced populations have not been addressed.

Author Response

See attached, please

Round 2

Reviewer 1 Report

Comments and Suggestions for Authors

Dear authors,

Your work in improving the manuscript clarity and quality was notable. I appreciate your effort. 

Reviewer 3 Report

Comments and Suggestions for Authors

Dear authors,

I would like to congratulate the authors for their efforts.

Best regards.